# A Fusion Transformer for Multivariable Time Series Forecasting: The Mooney Viscosity Prediction Case

**DOI:** 10.3390/e24040528

**Published:** 2022-04-09

**Authors:** Ye Yang, Jiangang Lu

**Affiliations:** 1State Key Laboratory of Industrial Control Technology, College of Control Science and Engineering, Zhejiang University, Hangzhou 310027, China; 11832016@zju.edu.cn; 2Zhejiang Laboratory, Hangzhou 311121, China

**Keywords:** multivariate time series forecasting, deep learning, positional encoding, static covariates, transformer, Mooney viscosity

## Abstract

Multivariable time series forecasting is an important topic of machine learning, and it frequently involves a complex mix of inputs, including static covariates and exogenous time series input. A targeted investigation of this input data is critical for improving prediction performance. In this paper, we propose the fusion transformer (FusFormer), a transformer-based model for forecasting time series data, whose framework fuses various computation modules for time series input and static covariates. To be more precise, the model calculation consists of two parallel stages. First, it employs a temporal encoder–decoder framework for extracting dynamic temporal features from time series data input, which analyzes and integrates the relative position information of sequence elements into the attention mechanism. Simultaneously, the static covariates are fed to the static enrichment module, which is inspired by gated linear units, to suppress irrelevant information and control the extent of nonlinear processing. Finally, the prediction results are calculated by fusing the outputs of the above two stages. Using Mooney viscosity forecasting as a case study, we demonstrate considerable forecasting performance improvements over existing methodologies and verify the effectiveness of each component of FusFormer via ablation analysis, and an interpretability use case is conducted to visualize temporal patterns of time series. The experimental results prove that FusFormer can achieve accurate Mooney viscosity prediction and improve the efficiency of the tire production process.

## 1. Introduction

Multivariable time series data are ubiquitous in a wide variety of domains, including financial analysis [1], medical analysis [2], weather forecasting [3], and renewable energy production [4]. Forecasting is one of the most sought-after tasks in analyzing time series data due to its importance in industrial, social, and scientific applications. There are numerous classical approaches for solving time series forecasting problems [5], including auto-regressive integrated moving average (Box–Jenkins ARIMA) models [6], exponential smoothing [7], and Kalman filter [8]. They incorporate prior knowledge about time series structures such as trend, seasonality, and other temporal patterns, and are capable of producing high-quality predictions for single linear time series. However, they are ineffective in predicting complex time series data, partly because of their inability to utilize the time-related features [9].

To tackle the aforementioned challenges in solving the modern large-scale multivariate time series forecasting problems, deep neural networks have been applied to model complicated sequential data. Naturally, complex patterns are mined for time series trends using deep learning models based on convolutional neural networks (CNN) [10] and recurrent neural networks (RNN) [11]. However, since the current hidden state of the RNN is dependent on the previous hidden state, the RNN’s forward calculation cannot be parallelized, limiting its computing efficiency. In addition, RNN has internal memory to process sequence data, and it suffers from gradient vanishing and exploding problems when processing long sequences [12]. Long short-term memory (LSTM) networks were specifically developed to address this limitation. LSTM employs three gates, an input gate, a forget gate, and an output gate, to modulate the information flow across the cells and alleviate gradient vanishing and exploding [13,14].

Sequence-to-sequence (Seq2Seq) [15] architecture was developed for machine learning applications and is now being applied to challenges of time series forecasting. It is comprised of three components, including an encoder, an intermediate vector, and a decoder. The encoder is a stack of LSTM or other recurrent units. Each unit accepts a single element from the input sequence. The intermediate vector is referred to as the context vector, which encodes all of the information from the input data. The decoder, like the encoder, is constructed using a stack of recurrent units and begins with the encoder vector as its first hidden state. Each recurrent unit generates an output element by computing its own hidden state. Seq2Seq has been widely applied in language translation tasks. However, its performance diminishes with time due to the inability of the intermediate vector to adequately encode a long sequence.

Transformer [16] was recently proposed as a method for processing a sequence of data that makes use of the attention mechanism. Transformer, in contrast to RNN-based approaches, does not analyze input in an ordered sequence. Instead, it analyzes the complete sequence of input and use self-attention mechanisms [17] to learn the sequence’s temporal relationships, potentially making it more suited to understanding recurrent patterns with long-term dependencies. Therefore, transformer-based models have the ability to model complex dynamics of time series data that are challenging for sequence models. There are few research results on using transformer-based models to solve multivariate time series forecasting problems, and the research content is primarily concerned with adapting the original transformer models used in natural language processing (NLP) to solve time series forecasting problems with minor modifications. Additionally, the effectiveness of these models is primarily evaluated using standard time series datasets, and there are few successful cases of practical applications in real-world scenarios.

The tire industry [18] has been developing rapidly in recent years, and the primary objective is to improve the quality of rubber. Mooney viscosity [19] is used to assess the quality of manufactured rubber, and this measurement determines whether the physical properties of tires meet the standards of practical use. The variables impacting the Mooney viscosity are classified as dynamic time series variables that reflect changes in the working conditions (e.g., temperature, pressure, etc.) and static covariates that represent the initial state of the working conditions (e.g., equipment number, raw material origin, etc.) [20]. At present, the majority of methods for determining the Mooney viscosity include placing the rubber in a specific machine and estimating the Mooney viscosity using the motor’s current [21]. There is, however, a time delay between the completion of rubber processing and the determination of its Mooney viscosity. If Mooney viscosity can be determined immediately after rubber refining, it will be critical for optimizing the process flow. As a result, Mooney viscosity prediction becomes a worthwhile subject of study.

Mooney viscosity forecasting can be considered as a type of multivariate time series forecasting issue, in which different time series variables at the manufacturing site are used to predict the rubber’s Mooney viscosity. For genuine rubber refining [22], the production process is divided into different stages, with each stage requiring a unique processing operation, as reflected in the temporal characteristics of dynamic time series components that are constantly changing at different stages. As a result, it is vital to capture and assess the time relationship between the various processing stages, particularly their relative order, which has a significant effect on the final predicted Mooney viscosity. Additionally, there are several static covariates in rubber refining that are scalar in nature and do not exhibit temporal dynamics, and some of them are completely irrelevant to the Mooney viscosity. Feeding them directly into the transformer model for nonlinear processing does not always result in optimal calculation results, but rather raises the computational cost. The Mooney viscosity prediction problem is merely a microcosm of the current challenges associated with solving complex time series prediction problems in real-world industrial scenarios. Therefore, it is critical to investigate a temporal fusion model that can handle time series input and static covariates differently, which can fully capture the time series input’s temporal dynamics and perform flexible nonlinear processing for static covariates, so as to achieve accurate prediction.

In this work, we use Mooney viscosity prediction as a case study to demonstrate that our proposed transformer-based model, FusFormer, can be successfully applied to the practical task of multivariable times series forecasting and outperforms existing forecasting approaches. Specifically, our contributions are the following:We develop a general transformer-based model for time series forecasting that is capable of performing simultaneous feature analysis on different types of variables and accurately predicting the target value.We propose a temporal encoder–decoder framework to cope with time series input. To be more precise, this framework introduces the concept of directed graphs to account for the relative positions information between the input time series elements and modifies the self-attention mechanism to better capture the temporal dynamics of the time series input.We proposed a static enrichment module, which is used to integrate information from static covariates. The gated linear unit (GLU) [23] is added in this module to filter out unnecessary information and to control the extent of nonlinear processing of static covariates, hence boosting the model’s flexibility to capture static features.We demonstrate, using the Mooney viscosity prediction as an experiment, that our proposed model is capable of producing state-of-the-art forecasting results. Furthermore, we provide an interpretable use case of the model by visualizing average importance weight distribution of time steps.

The rest of this paper is organized as follows. Section 2 briefly reviews related works on time series forecasting. Section 3 proposes the background of our proposed model. Section 4 describes the architecture and mathematical principles of the proposed model. In Section 5, we demonstrate the effectiveness of the proposed model in the Mooney prediction case. Section 6 concludes this paper and discusses future work.

## 2. Related Work

### 2.1. Time Series Forecasting

Time series forecasting has demonstrated its wide applications in business and industrial decision masking. Previously, the majority of time series forecasting literature has relied heavily on statistical models. The Box–Jenkins ARIMA family of methods develop a model in which the prediction is a weighted linear sum of recent past data or lags. Liu et al. [24] applied online learning to ARIMA models for time series forecasting. The matrix factorization methods [25,26] model related series data as a matrix and attempt to learn information across the time series. However, it is difficult to predict the modern time series by these traditional statistical models because of their linear assumption and restricted scalability.

In recent years, deep neural networks, particularly recurrent neural networks capable of dealing with time series data, have been proposed to solve time series forecasting. Khedhiri [14] compared the seasonal autoregressive fractionally integrated moving average (SARFIMA) and LSTM on a temperature prediction task, illustrating the better performance of deep learning models. Wan et al. [27] proposed a convolutional LSTM encoder–decoder network to achieve accurate forecasting in non-periodic datasets. Salinas et al. [28] proposed DeepAR, which uses stacked LSTM layers to generate parameters of one-step-ahead Gaussian predictive distributions. Rangapuram et al. [29] proposed deep state-space models (DSSM), which adopt a similar approach, utilizing LSTMs to generate parameters of a predefined linear state-space model with predictive distributions produced via Kalman filtering. Wen et al. [30] use an RNN as an encoder and multi-layer perceptrons (MLPs) as a decoder to solve the so-called error accumulation issue and conduct multi-ahead forecasting in parallel. Lai et al. [31] developed LSTNet, which is based on RNN architecture and is auto-regressive to catch temporal dependencies. Wen et al. [30] provided the multi-horizon quantile recurrent forecaster (MQRNN), which generates context vectors for each horizon using LSTM or convolutional encoders.

### 2.2. Transformer Models for Time Series

The well-known self-attention-based transformer has recently been proposed for sequence modeling and has achieved great development, including successful applications in translation, voice, music, and image generation. Several transformer-based models have been used for time series problems in the recent past. Li et al. [32] proposed LogSpare transformer to deal with time series problems, which shows superior performance compared to other deep learning models on four public forecasting datasets. Wu et al. [33] used a transformer for forecasting influenza prevalence and similarly showed performance benefits compared to ARIMA and LSTM. Lim et al. [34] used a transformer for multi-horizon univariate forecasting that uses gating mechanisms to restrict variables that are not relevant to the prediction target. Wu et al. [35] propose an adversarial sparse transformer (AST), which adopts a sparse transformer as the generator to learn a sparse attention map for time series forecasting, and uses a discriminator to improve the prediction performance at a sequence level.

## 3. Background

### 3.1. Rubber Refining Process

The production process of rubber refining [36] is mainly divided into three stages: wetting, dispersing, and kneading. The main task of the wetting stage is to load the production materials and raw rubber into the compacting machine for preliminary refining, with the refining primarily manifested in the mixing of the raw rubber and carbon black into a single mass; the main task of the dispersion stage is to make the rotor rotate at a high speed. The rubber-carbon black aggregates are cut and pulverized, dispersed into the raw rubber, and gradually form a bonded rubber; the kneading step is responsible for further mixing and adjusting the rubber’s Mooney viscosity, which is a critical stage in determining the final Mooney viscosity.

### 3.2. Problem Description

Assume that a given time series dataset collection contains *I* unique samples, such as separate production records from the rubber refining process. Each sample *i* is associated with a set of static covariates $s_{i}$, as well as time series input sets $x_{i},i=1,\cdots ,t_{N}$ and scalar targets $y_{i}$. The Mooney viscosity prediction takes the form:(1)$${\hat{y}}_{i}=f\left(x_{i},s_{i}\right)$$
where ${\hat{y}}_{i}$ is the predicted Mooney viscosity, and f(.) is the proposed FusFormer in this work. Each time series input $x_{i}$ is composed of different time series of length $t_{N}$, whereas the static covariates set $s_{i}$ contains a variety of scalar features. The Mooney viscosity prediction can be considered as a supervised multivariable time series prediction task.

### 3.3. Transformer

Transformer [16] is a novel encoder–decoder model that is based on the attention mechanism and totally removes recurrent neural networks, which can compute the sequence effectively. The canonical transformer employs an encoder–decoder structure, consisting of stacked encoder and decoder layers. As shown in Figure 1, encoder layers are composed of two sublayers: self-attention, followed by a position-wise feed-forward layer. The encoder produces a vector to feed to the decoder. Decoder layers consist of three sublayers, which are followed by encoder–decoder attention, and a position-wise feed-forward layer. The decoder uses masking in its self-attention to prevent a given output position from incorporating information about future output positions during training. Transformer accelerates training speed and convergence by utilizing residual connections surrounding each of the sublayers, followed by layer normalization.

Different from batch normalization, layer normalization estimates the normalized statistics from the sum of inputs to the neurons within the hidden layer, which does not introduce any new dependencies during the training process. It has a beneficial effect on RNN and can improve the generalization ability of the model. In addition, it has also been used in transformer-based models.

Positional encoding [37] based on sinusoids of varying frequency are added to encoder and decoder input elements prior to the first layer. In contrast to learnable or absolute position representations, the authors hypothesized that sinusoidal position encodings would help the model to generalize to sequence lengths unseen during training. The formula for representing the position of the elements in a time series using the sine and cosine functions is as follows:(2)$$\begin{array}{c} PE\left(pos,2i\right)=\sin\left(\frac{pos}{10^{8i/d_{model\;}}}\right)\\ PE\left(pos,2i+1\right)=\cos\left(\frac{pos}{10^{8i/d_{model\;}}}\right) \end{array}$$
where pos is the position, and *i* is the dimension. That is, each dimension of the positional encoding corresponds to a sinusoid.

Self-attention sublayers employ multiple attention heads. To form the sublayer output, results from each head are concatenated and then transformed using a parameterized linear transformation. Each attention head operates on an input sequence $x=\left(x_{1},\ldots ,x_{n}\right)$ of *n* elements, and computes a new sequence $z=\left(z_{1},\ldots ,z_{n}\right)$ of the same length. Each output element $z_{i}$ is computed as weighted sum of linearly transformed input elements:(3)$$z_{i}=\sum_{j=1}^{n}\alpha_{ij}\left(x_{j}W^{V}\right)$$

Each weight coefficient $\alpha_{ij}$ is computed using a softmax function:(4)$$\alpha_{ij}=\frac{e_{ij}}{\sum_{k=1}^{n}\exp e_{ik}}$$

In addition, $e_{ij}$ is computed using a compatibility function that compares two input elements:(5)$$e_{ij}=\frac{\left(x_{i}W^{Q}\right){\left(x_{j}W^{K}\right)}^{T}}{\sqrt{d_{z}}}$$
where $W^{Q},W^{K},W^{V}$ are parameter matrices and $xW^{Q}$, $xW^{K}$, $xW^{V}$ generate three vectors *Q*, *K*, *V*, which are abstractions conducive to the calculation of the self-attention. Simply put, *Q* represents the query vector, *K* represents the vector of the correlation between the queried information and contextual information, and *V* represents the queried vector. Scaled dot product was chosen for the compatibility function, which calculates the similarity of two elements and enables efficient computation.

## 4. Methodology

Many time series forecasting problems require modeling different types of inputs to achieve more accurate predictions, involving the capture of temporal dynamics of time series input as well as the nonlinear processing of static covariates. To address the above challenges, we propose the FusFormer.

In general, the FusFormer is composed of a temporal encoder–decoder framework and an auxiliary static enrichment module, as demonstrated in Figure 2. The original input fed to the model will be divided into two parts, which are time series input and static covariates input. The time series input are pre-processed by the time series data processing scheme and fed to the temporal encoder–decoder framework for extracting dynamic temporal features, while the static covariates are fed to the static enrichment module to give the model the flexibility to apply non-linear processing only where needed. The final prediction result is calculated by fusing the outputs of the two parallel stages described above.

The major constituents of our proposed model are:**Time series data-processing scheme** for mapping the input time series data into a vector of given dimensions, while also learning the local context information in the time series, and finally encoding the position of the time series input.**Temporal encoder–decoder framework** for taking into account the relative position information of the input time series elements, incorporating relative position representations into the self-attention mechanism, and lastly calculating a temporary vector comprising temporal dynamic.**Static enrichment module** for incorporating static characteristics of static covariates into the model, using GLU to suppress the irrelevant information and control the extent of nonlinear processing to improve the prediction accuracy.

### 4.1. Time Series Data-Processing Scheme

The time series data-processing scheme is developed to perform targeted preprocessing on time series data, which is beneficial to transformer to better extract the input time series temporal dynamics. As shown in Figure 3, it mainly includes three components: linear projection, local feature extraction, and positional encoding.

Once the time series input is fed to FusFormer, the linear projection layer maps the data to a vector of the specified dimension using a fully connected network. This step is essential for the model to employ a multi-head attention mechanism.

Temporal patterns in time series data may evolve with time significantly due to various events, such as wetting and kneading operations in the rubber refining process. Thus, whether an observed point in a time series is an outlier, a change point, or a part of the pattern is very context dependent. However, in the self-attention layers of transformer, the similarities between queries and keys are computed solely on the basis of their point-wise values, without fully exploiting local context information, as shown in Figure 4a. Agnosticism of local context may confuse the self-attention module in terms of whether the observed point is a change point or a component of patterns, resulting in inferior prediction results [32].

To ease this issue, we employ dilated causal convolution [38] to transform time series input data with proper paddings. Notably, causal convolution is a type of convolution used for temporal data that ensures the model cannot violate the ordering in which we model the data. As illustrated in Figure 4b, by employing causal convolution, the self-attention calculation in transformer can be more aware of local context; therefore, it can compute attention scores using information about their local shapes rather than point-wise values. However, since the rubber refining process is complex and the length of sequence data collected is long, it is necessary to enlarge the reception field of causal convolution on time series to extract more contextual history information. Therefore, we introduce the dilated casual convolution to address above-mentioned issue. The dilated causal convolution is a kind of causal convolution where the filter is applied over an area larger than its length by skipping input values with a certain step, which effectively allows the network to have very large receptive fields with just a few layers.

More formally, for a time series input *X* and a filter f:{0,…,k−1}→R, the dilated casual convolution operation *F* on element *x* of the time series corresponding to moment *t* is defined as follows:(6)$$F\left(x\right)=\sum_{i=0}^{k-1}f\left(i\right)\cdot x_{t-d\cdot i}$$
where *d* is the dilation factor, *k* is the filter size, and t−d·i accounts for the direction of the past. Dilation is thus equivalent to introducing a fixed step between every two adjacent filter taps. When d=1, a dilated casual convolution reduces to a regular casual convolution. Using larger dilation enables the local enhancement layer to represent a broader range of inputs, effectively enlarging the receptive field of the FusFormer to characterize longer-term input and aiding in prediction performance improvement.

Finally, as with the transformer, fixed positional encoding with sine and cosine functions is used to encode sequential information in the time series data by element-wise addition of the input vector with a positional encoding vector.

### 4.2. Temporal Encoder–Decoder Framework

The main components of transformer are the encoder and decoder, both of which contain self-attention sub-layers and fully connected feed-forward sub-layers, contributing to the great success of transformer in natural language processing. In order to solve the multivariate time series forecasting problem, as shown in Figure 5, we optimized the original encoder–decoder architecture by introducing a relative position-aware layer and incorporating relative position representation into the self-attention mechanism, and effectively integrating them into the temporal encoder–decoder framework.

#### 4.2.1. Relative Position-Aware Layer

Transformer completely relies on the self-attention mechanism and has achieved the most advanced performance in machine translation. In comparison to RNN and CNN, it does not explicitly model position information in the structure; instead, it employs positional encoding to address the aforementioned issues. However, when it comes to time series forecasting, time series data in many real-world circumstances exhibits phased variations, which frequently have a non-negligible effect on the prediction. As a result, we consider the relative position information of elements in order to extract more temporal dynamics of time series data.

In the canonical transformer, each element in the time series has a fixed position vector determined by Equation (Equation 2). The input of the transformer is the addition of the time series input and its positional encoding vector. The calculation step related to position encoding in transformer is self-attention. Take the calculation of the attention score of any two elements $x_{i}$ and $x_{j}$ in time series as an example:(7)$$\begin{array}{cc} Attn_{i,j} & ={\left(W_{q}\left(x_{i}+PE_{i}\right)\right)}^{T}\left(W_{k}\left(x_{j}+PE_{j}\right)\right)\\ \; & =\underset{\left(a\right)}{\underset{︸}{x_{i}^{⊤}W_{q}^{⊤}W_{k}x_{j}}}+\underset{\left(b\right)}{\underset{︸}{x_{i}^{⊤}W_{q}^{⊤}W_{k}PE_{j}}}+\underset{\left(c\right)}{\underset{︸}{PE_{i}^{⊤}W_{q}^{⊤}W_{k}x_{j}}}+\underset{\left(d\right)}{\underset{︸}{PE_{i}^{⊤}W_{q}^{⊤}W_{k}PE_{j}}} \end{array}$$

Equation (Equation 7) consists of four parts: the term (a) does not contain position information; terms (b) and (c) have a single position vector, so they do not contain relative position information; and (d) contains both $PE_{i}$ and $PE_{j}$, and is likely to contain relative position information. In fact, according to the position encoding method (7), if there is no matrix $W_{q}$ and $W_{k}$ in term (d) of Equation (Equation 7), then it contains relative position information, since the positional coding of the $t_{th}$ element in Equation (Equation 2) in the time series is expressed as follows:(8)$$PE_{t}=\left[\begin{array}{c} \sin\left(m_{0}t\right)\\ \cos\left(m_{0}t\right)\\ \vdots \\ \sin\left(m_{\frac{d_{model}}{2}-1}t\right)\\ \cos\left(m_{\frac{d_{model}}{2}-1}t\right) \end{array}\right]$$
where $d_{model}$ is the encoding dimension, and $m_{i}$ is the constant whose value is $1/10,000^{2i/d_{model}}$. Therefore,
(9)$$\begin{array}{cc} PE_{t}^{T}PE_{t+k} & =\sum_{j=0}^{\frac{d_{model}}{2}-1}\left[\sin\left(m_{j}t\right)\sin\left(m_{j}\left(t+k\right)\right)\right.\left.+\cos\left(m_{j}t\right)\cos\left(m_{j}\left(t+k\right)\right)\right]\\ \; & =\sum_{j=0}^{\frac{d_{model}}{2}-1}\cos\left(m_{j}\left(t-\left(t+k\right)\right)\right)\\ \; & =\sum_{j=0}^{\frac{d_{model}}{2}-1}\cos\left(m_{j}k\right) \end{array}$$

It can be seen that the final result is only related to the relative position of the two positions, that is, Equation (Equation 9) contains relative position information. However, when the matrix is included in Equation (Equation 9), as in term (d), the relative position information cannot be reflected clearly. To prove this claim, the two formulas $PE_{t}^{T}PE_{t+k}$ and $PE_{t}^{T}WPE_{t+k}$ are visualized in Figure 6. $d_{model}$ is set to be 128, and *W* in $PE_{t}^{T}WPE_{t+k}$ is the trained matrix in transformer. As illustrated in Figure 6, the above curve corresponds to $PE_{t}^{T}PE_{t+k}$, with obvious trends and patterns, which means that it can reflect relative position information. In contrast, the lower curve has no clear pattern. Therefore, it can be concluded that the self-attention calculation will lead to the loss of relative position information.

Motivated by the above findings, relative position information needs to be specifically added to the self-attention computation. As illustrated in Figure 7, to represent the relative position of elements in the time series data, we model the time series data as a directed, fully connected graph [39]. Suppose there are two elements in the time series denoted as $x_{i}$ and $x_{j}$, which are connected by two opposite-direction edges $edge_{ij}^{V}$ and $edge_{ij}^{K}$, which are used to represent the relative position information between the two elements. In addition, we assume that once the distance between two elements in the time series reaches a certain value, their relative position information is insufficient to have a substantial effect on the prediction results. As a result, the greatest effective distance between elements is limited to *k*, and 2k+1 unique edges label are considered:(10)$$\begin{array}{cc} edge_{ij}^{K} & =w_{\mathrm{clip}(j-i,k)}^{K}\\ edge_{ij}^{V} & =w_{\mathrm{clip}(j-i,k)}^{V}\\ \mathrm{clip}(x,k) & =\max(-k,\min(k,x)) \end{array}$$
where $w^{V}=\left(w_{-k}^{V},\ldots ,w_{k}^{V}\right)$ and $w^{K}=\left(w_{-k}^{K},\ldots ,w_{k}^{K}\right)$ are learnable parameters, $w_{i}^{V},w_{i}^{K}\in R^{d_{model}}$; these two matrices are developed to represent relative position information of elements in the time series data.

#### 4.2.2. Relative Multi-Head Attention

Apart from effectively capturing the relative position information between elements, another motivation for learning two distinct edge representations is that $edge_{ij}^{V}$ and $edge_{ij}^{K}$ are suitable for modifying Equations (3) and (5) in self-attention, respectively, without requiring additional linear transformations. These representations can be shared across attention heads. Equation (Equation 3) is modified to propagate edge information to the sublayer output:(11)$$z_{i}=\sum_{j=1}^{n}\alpha_{ij}\left(x_{j}W^{V}+edge_{ij}^{V}\right)$$

This modification is presumably important for tasks in which relative position information represented by the edge selected by a given attention head is useful to downstream encoder or decoder layers. Then, Equation (Equation 5) is modified to consider edges when determining compatibility:(12)$$e_{ij}=\frac{x_{i}W^{Q}{\left(x_{j}W^{K}+edge_{ij}^{K}\right)}^{T}}{\sqrt{d_{z}}}$$

The advantage of using simple addition to incorporate edge representations in Equations (7) and (8) is that it enables an efficient implementation. Transformer computes self-attention efficiently for all sequences, heads, and positions in a batch using parallel matrix multiplication operations [16]. Without employing relative position representations, each $e_{ij}$ can be determined by performing parallel matrix multiplications, as shown in Equation (Equation 4). For any sequence and head, this requires sharing the same representation for each position across dot product function with other positions.

We discover that the representation of relative positions varies across pairs of positions, which prevents us from computing all $e_{ij}$ for all pairs of positions in a single matrix multiplication. To resolve the above-mentioned issue, we split the computation of Equation (Equation 8) into two terms:(13)$$e_{ij}=\frac{x_{i}W^{Q}{\left(x_{j}W^{K}\right)}^{T}+x_{i}W^{Q}{\left(edge_{ij}^{K}\right)}^{T}}{\sqrt{d_{z}}}$$

The first term is identical to Equation (Equation 4), which can be computed efficiently. For the second term involving relative position representation, we reshape the $edge_{ij}^{K}$ to match the matrix dimension of $x_{i}W^{Q}$, allowing us to successfully apply matrix multiplication to compute the second term. The final $e_{ij}$ can be calculated by reshaping two terms and adding them together. The same approach can be used to efficiently compute Equation (Equation 7), with the following modification:(14)$$z_{i}=\sum_{j=1}^{n}\left(\alpha_{ij}\left(x_{j}W^{V}\right)+\alpha_{ij}\left(edge_{ij}^{V}\right)\right)$$

In general, to improve the learning capacity of the relative self-attention, relative multi-head attention is introduced, employing different heads for different representation subspaces:(15)$$\mathrm{multi-head}\left(Q,K,V\right)=\left[H_{1},\ldots ,H_{m_{H}}\right]W_{H}$$
(16)$$H_{h}=Attention\left(QW_{h}^{Q},KW_{h}^{K},VW_{h}^{V},edge^{V},edge^{K}\right)$$
where Attention(.) is the the relative self-attention function.

#### 4.2.3. Optimized Encoder Layer

The optimized encoder layer (opt-encoder layer) in FusFormer is structurally similar to the encoder layer in the transformer. The primary distinction is that the proposed opt-encoder layer incorporates both the relative position-aware layer and relative multi-head attention.

Each opt-encoder layer is composed of two sub-layers: a relative multi-head attention sub-layer and a fully connected feed-forward sub-layer. Each sub-layer is followed by a normalization layer. The encoder generates a temporary vector to feed to the decoder. Taking the calculated output of the time series data-processing scheme as opt-encoder layer input, the specific implementation steps are as follows:Input data processed by the time series data-processing scheme is fed to the opt-encoder layer.A relative position-aware layer is utilized to extract the sequence-relative position information from input data of step1.The output of step2 is passed to the relative multi-head attention layer; then, the addition and layer normalization operations are performed.The result of step3 is fed to the position-wise feed-forward layer, and the addition and layer normalization operations are performed once more to compute the final output, which is used as part of the input to the opt-decoder layer.

#### 4.2.4. Optimized Decoder Layer

We employ an optimized decoder layer (opt-decoder layer) design that is analogous to the canonical transformer architecture. In addition to the two sub-layers contained inside each opt-encoder layer, the opt-decoder layer incorporates a third sub-layer to apply relative multi-head attention over the opt-encoder output. Taking the calculated output of the opt-encoder layer and the last time step slice of the encoder input as opt-decoder layer input, the specific implementation steps are as follows:The opt-decoder layer input is fed to the original multi-head attention, and the addition and layer normalization operations are performed.The output of step1 is fed to the relative position-aware layer, which captures the sequence-relative position information.The output of step2 and the opt-encoder layer output are used as the input of the relative multi-head attention, and the addition and layer normalization operations are performed.The result of step3 is fed to the position-wise feed-forward layer, which performs the addition and layer normalization operations once more to compute the final opt-decoder layer output.

### 4.3. Static Enrichment Module

Along with the time series input, the rubber refining process also collects static covariates that do not contain temporal dynamics. The precise relationship between these exogenous static covariates and Mooney viscosity is often unknown in advance, making it difficult to anticipate which variable is relevant. In previous research [16,32,33], all static covariates are used as input for nonlinear calculations without discrimination. However, it is possible that irrelevant information in static covariates will have a negative effect on the prediction performance. Additionally, it is difficult to determine the extent of the required nonlinear processing. Too many nonlinear layers in a neural network may result in overfitting, and there may be instances where simpler models can be beneficial.

With the motivation of giving the model the flexibility to apply nonlinear processing only when necessary, we propose the static enrichment module (SEM) shown in Figure 8 as a building block of our proposed method. The SEM takes in the static covariates *a* and yields:(17)$$hid_{1}=\mathrm{ELU}\left(W_{1}a+b_{1}\right)$$
(18)$$hid_{2}=W_{2}hid_{1}+b_{2}$$
(19)$$Out_{GLU}=\sigma \left(W_{3}hid_{2}+b_{3}\right)⊙\left(W_{4}hid_{2}+b_{4}\right)$$
(20)$$SEM\left(a\right)=\;\mathrm{LayerNorm}\;\left(a+Out_{GLU}\right)$$
where σ(.) is the sigmoid activation function, $W_{\left(.\right)}$ and $b_{\left(.\right)}$ are the weights and biases, ⊙ is the element-wise Hadamard product, $hid_{1}$ and $hid_{2}$ are intermediate states, and LayerNorm is standard layer normalization. Equation (Equation 19) is the component gating layer based on the gated linear unit (GLU) [28], which is introduced to enhance the flexibility of performing nonlinear processing on the static covariates’ input. This approach is inspired by the LSTM’s gating mechanism, which employs forget gates for selective ignoring of past information. The sigmoid function has a value range of 0 to 1, which acts as a soft feature selection function. When certain static covariates are not related to the prediction target during the training process, the corresponding sigmoid output value will be close to 0, implying that nonlinear processing is suppressed. Therefore, GLU allows the SEM module to control the extent of nonlinear processing of the static covariates’ input.

As illustrated in Equation (Equation 17), ELU is the exponential linear unit activation function [34], and its formula is as follows:(21)$$\mathrm{ELU}\left(x\right)=\left\{\begin{array}{cc} x & \;\mathrm{if}\;x>0\\ \beta (e^{x}-1) & \;\mathrm{if}\;x\leq 0 \end{array}\right.$$
when $W_{1}a+b_{1}>>0$, the ELU activation would act as an identity function, and when $W_{1}a+b_{1}<<0$, the ELU activation would generate a constant output. The ELU function is continuous and differentiable at all points, and it has a saturation region in the negative domain, which is robust to noise.

The output of SEM and the output of the temporal encoder–decoder framework are spliced together to generate a temporary vector that is then passed to the fully connected layer to calculate the final predicted value.

## 5. Experiment

### 5.1. Dataset and Evaluation Index

We use the industrial data collected from the rubber refining production site of a local rubber company as the data set. We filter the raw data in advance, removing any samples data that do not meet the standard process criteria, and finally reserve 200,000 samples for evaluating the algorithm’s prediction performance. Each sample in this data set contains nine variables, among which the dynamic variables in the form of time series include temp, speed, power, and press; the static covariates in the form of scalar include seconds, energy, equip, type1, and type2; and the prediction target is Mooney viscosity. The following is a detailed description of the variables discussed previously:
**Temp** is used to record temperature changes of the production site.**Speed** is used to record the change of the rotor speed of the compacting machine.**Power** is used to record the change of the power of the compacting machine.**Press** is used to record the pressure change of the upper top bolt of the compacting machine.**Seconds** is used to record total rubber refining time.**Energy** is used to record cumulative total energy of rubber refining.**Equip** is used to indicate the equipment number of the compacting machine.**Type1** is used to indicate the main category number of the produced rubber.**Type2** is used to indicate the sub-category number of the produced rubber.

We perform min-max scaling and label encoding on the dataset and divide it into training set, validation set, and test set; the ratio of division is 7:2:1. Then, we introduce the root mean square error (RMSE) and root relative squared error (RRSE) as the evaluation index between the actual Mooney viscosity $y_{true}$ and the predicted Mooney viscosity $y_{pred}$, which is expressed as follows:(22)$$RMSE=\sqrt{\frac{1}{m}\sum_{i=1}^{m}{\left(y_{true}-y_{pred}\right)}^{2}}$$
(23)$$RRSE=\sqrt{\frac{\sum_{i=1}^{m}{\left(y_{true}-y_{pred}\right)}^{2}}{\sum_{i=1}^{m}{\left(y_{true}-\bar{y_{true}}\right)}^{2}}}$$
where $\bar{y_{true}}$ represents the average values of $y_{true}$.

### 5.2. Training Procedure

We partition all time series into three parts—a training set for learning, a validation set for hyperparameters tuning, and a hold-out test set for performance evaluation. Hyperparameters optimization is accomplished via random search, using 60 iterations for our data set. Full search ranges for all hyperparameters are given, with the optimal model parameters listed in Table 1.

*Model dimension*—64, 128, 256, 512;*Encoder layer num*—1, 2, 3, 4;*Decoder layer num*—1, 2, 3, 4;*Dropout rate*—0.01, 0.02, 0.05, 0.1, 0.2;*Batch size*—16, 32, 64, 128;*Learning rate*—0.00001, 0.0001, 0.001;*Kernel size*—1, 2, 3, 4;*Dilation Factor*—1, 2, 3, 4;*Clip distance*—40, 50, 60, 70.

Our model is trained on a workstation with Nvidia RTX 3090 GPU and Intel Xeon-E5 2.50GHz CPU, and it can be deployed without the need for extensive computing resources.

### 5.3. Comparison Methodologies

We extensively compare FusFormer to a wide range of models for Mooney viscosity forecasting. This subsection describes other models to benchmark our proposed model.

**LightGBM** [40]: The LightGBM (LGB) is a model trained based on the histogram decision tree. LGB takes all variables as input, and the outputs predict Mooney viscosity after iterative training. Parameters numleaves and maxdepth are set to be 75 and 6, respectively. For training, the RMSE evaluation metric, L1 regularization, and L2 regularization are utilized.

**LSTM** [41]: The LSTM model has a stack of two LSTM layers and a final dense layer to predict the Mooney viscosity directly. The LSTM layers encode sequential information from the input through the recurrent network, and the dense connected layer takes the final output. The two LSTM layers are 128 and 64 units, respectively. A dropout rate of 0.1 is applied to LSTM layers for regularization. Adam optimizer and a learning rate of 0.02 are used for training.

**Residual Regression Model** [42]: The deep residual neural network (RseNet1D) is a modified Resnet, which replaces convolutional layers and pooling layers with fully connected layers. It has a stack of dense blocks and identity blocks and a final dense layer. The input dimension is set to be 64, and the number of dense blocks and identity blocks is 3. Adam optimizer and a learning rate of 0.001 are used for training.

**Transformer** [16]: The canonical transformer employs an encoder–decoder structure, consisting of stacked encoder and decoder layers. All types of variables are fed to the input layer, the context vector is generated by the encoder, and the decoder followed by a dense layer finally outputs the prediction result. The hyperparameters of the canonical transformer are identical to those of our proposed model.

**Convolutional Transformer** [32]: The convolutional transformer (conv-transformer) is a transformer-based model that incorporates CNN to generate changed queries and keys, hence rendering the model susceptible to time series anomalies. The calculation process of Conv-Transformer is the same as that of canonical transformer. The kernel size is 3, and other hyperparameters used for training are identical to those used in the canonical transformer.

### 5.4. Results and Discussion

In this experiment, we test six algorithms, including the proposed FusFormer, and evaluate the prediction performance of each method by calculating the RMSE between the predicted Mooney viscosity and actual Mooney viscosity.

As shown in Table 2, we evaluate the prediction performance of the FusFormer to that of LSTM, LightGBM, ResNet1D, Transformer, and Conv-Transformer. The RMSE and RRSE for each approach, as well as the relative performance improvement over LSTM, are summarized in Table 2. The comparison indicates that FusFormer outperforms other models in terms of forecasting performance.

LSTM has the lowest prediction accuracy of all approaches, owing to its inability to deal with long time series. LightGBM and ResNet1D perform similarly, and their prediction results are more accurate than those of LSTM. LightGBM, as a tree model, is good at obtaining nonlinear relationships in the data and identifying significant features, allowing for accurate prediction results. On the other hand, ResNet1D, as a deep network composed of numerous convolutional and fully connected layers, is capable of both nonlinear approximation and accurate prediction.

In terms of transformer-based models, transformer and conv-transformer have better prediction results than LightGBM and ResNet1D, reflecting the effectiveness of the self-attention mechanism in transformer. Considering that conv-transformer introduces CNN to enhance the analysis of local information of time series, its prediction effect is slightly better than that of transformer.

Compared with the preceding transformer-based model, FusFormer has a significant improvement in prediction results, and the RMSE of FusFormer is reduced by 22.48% and 20.09%, respectively. The improved prediction results are due to FusFormer’s ability to better capture the temporal dynamics with relative position information learning, and it also has a static enrichment module for flexible nonlinear processing of static covariates, which is not available in the other two transformer-based models.

To intuitively reflect the prediction accuracy of our FusFormer, we display the prediction results of FusFormer and conv-transformer. It should be noted that the prediction curves in Figure 9 and Figure 10 are made up of 200 randomly picked prediction samples from the test set for the purpose of visualizing the prediction error clearly. As illustrated in Figure 10, FusFormer is capable of reliable prediction for the vast majority of samples, although there are a few samples with non-negligible errors between predicted and true values.

The prediction performance of conv-transformer is clearly inferior to FusFormer. There are non-negligible errors between majority predicted values and true values, and the coincidence of the true value curve and the predicted value curve is low, indicating the superiority of FusFormer’s prediction performance as well as the validity of the temporal encoder–decoder framework and static enrichment module.

### 5.5. Ablation Analysis

To quantify the benefits of each of our proposed architectural contribution, we conduct a detailed ablation analysis, removing or replacing critical components from the architectural as below, and quantifying the percentage increase in RMSE compared to transformer.

#### 5.5.1. Local Enhancement Layer (LEL)

To validate the effectiveness of the local enhancement layer, we delete this layer in the time series data-processing scheme, and compare the training loss curve and prediction results with FusFormer.

It can be seen from Figure 11a that the loss curve of the model after deleting the local enhancement layer has slower convergence, and the accuracy of prediction has also deteriorated, as indicated in Table 3. This further illustrates the effectiveness of the local enhancement layer and the importance of local context information extraction in the prediction process.

#### 5.5.2. Relative Position-Aware Layer and Relative Multi-Head Attention (RPL and RMA)

To demonstrate the effectiveness of the relative position-aware layer, as well as the relative multi-head attention, we ablate by deleting the relative position-aware layer and replacing the relative multi-head attention with the ordinary multi-head attention in each encoder layer and decoder layer, and compare the training loss curve and prediction results with FusFormer.

As illustrated in Figure 11b, after replacing the relative position-aware layer and the relative multi-head attention, the training loss values increase, indicating that the training performance is no longer as good as it was previously. This can also be reflected in the RMSE indicator in Table 3. The prediction accuracy is also lower than FusFormer. This comparison proves that analyzing the relative position information between time series elements is critical for capturing temporal dynamics and calculating more accurate prediction results.

#### 5.5.3. Static Enrichment Module (SEM)

To prove the superiority of the static enrichment module, we ablate by replacing this module with a simple feed-forward layer, and compare the training loss curve and prediction results with FusFormer.

As illustrated in Figure 11c, the training loss value reduces somewhat more slowly after the SEM module is replaced. The RMSE evaluation metrics in Table 3 show a modest reduction in prediction accuracy, which illustrates the superiority of the SEM module in dealing with static covariates. Additionally, we discover that FusFormer without the SEM module outperforms transformer in terms of prediction performance, demonstrating the superiority of the temporal encoder–decoder framework for handling time series data.

#### 5.5.4. Gated Linear Unit (GLU)

GLU is the core component in the SEM module. To verify the effectiveness of GLU in nonlinear processing, we ablate by replacing GLU with a simple linear layer, and compare the training loss curve and prediction results with FusFormer.

As illustrated in Figure 11d, after replacing the gated linear unit, the loss function curve fluctuates greatly in the first 20 epochs, and then it quickly converges to a stable value. In terms of prediction performance, the prediction accuracy after replacing the GLU with a linear layer has a small decrease, which reflects the effectiveness of the GLU in non-linear calculation.

Overall, all of the modules mentioned above contribute positively to FusFormer, notably by accelerating model training convergence and increasing prediction accuracy.

### 5.6. Hyperparameters Sensitivity Analysis

The hyperparameter sensitivity analysis of FusFormer is performed in the Mooney viscosity case under the univariate setting. We conduct sensitivity analysis on the main hyperparameters and then train the model using multiple values for each hyperparameter, comparing the training loss curve and RMSE for each value.

By combining Figure 12 and Table 4 below, we demonstrate that the hyperparameters of FusFormer are optimal. Simultaneously, the analysis results indicate that any modification in hyperparameters values has a varied influence on the training and prediction results. Hyperparameter *model dimension* has the greatest influence on prediction accuracy. A value that is either too little or too large will have a detrimental influence on the prediction performance, especially when the model dimension is 512, causing the training curve to converge slowly and reducing the model’s prediction accuracy. For the *Encoder layer num* and *Decoder layer num*, considering that the encoder input is a long sequence of data, a value of *Encoder layer num* that is too small will reduce prediction accuracy; while the decoder input is the last data point of the encoder input, a value of *Decoder layer num* that is too large will prevent training from convergence quickly. The different values of the remaining hyperparameters have relatively little influence on the prediction results.

### 5.7. Interpretability Use Case

Neural networks are frequently referred to as a kind of black box model, in large part because of their lack of feature representation in the training process. In a real rubber refining scenario, the influence of the different stages of production on the Mooney viscosity varies, and it is of practical importance to visualize the distribution of these influences in order to assist plants in optimizing their production processes.

#### 5.7.1. Interpretable Relative Multi-Head Attention

Given that the general multi-head attention in transformer uses different values for each head, attention weights alone would not be indicative of a particular feature’s importance. Inspired by the previous study by Google [31], we introduce interpretable relative multi-head attention to share values in each head, and employ additive aggregation of all heads:(24)$$\;\mathrm{Interpretable}\;\mathrm{multi-head}\;\left(Q,K,V\right)=\tilde{H}W_{H}$$
(25)$$\begin{array}{cc} \tilde{H} & =\left\{1/H\sum_{h=1}^{m_{H}}A\left(QW_{h}^{Q},KW_{h}^{K},edge^{K}\right)\right\}\left(VW_{V}+edge^{V}\right)\\ \; & =1/H\sum_{h=1}^{m_{H}}Attention\left(QW_{h}^{Q},KW_{h}^{K},VW^{V},edge^{K},edge^{V}\right) \end{array}$$
where A(.) is Equation (Equation 9), which calculates the similarity between query and key, $W_{V}$ is the value weights matrix shared across all heads, and $W_{H}$ is used for final linear mapping. According to Equation (Equation 25), we find that each head can learn different temporal patterns, while attending to a common set of time series input features, which can be viewed as a ensemble over attention weights.

Interpretable relative multi-head attention can be used to shed light on the important past time steps related to the Mooney viscosity. In contrast to other traditional and machine learning time series methods, which rely on model-based specifications for seasonality and lag analysis, FusFormer can learn such temporal patterns directly from raw time series data.

#### 5.7.2. Visualizing Attention Weight Patterns of Time Series

As shown in Figure 13, the curve is divided into four parts. The first part has no effect on the forecast since we filled the beginning of the time series with a value of 0 to provide a constant input length for the time series, and so this section lacks temporal patterns. The remaining three parts correlate to the three stages in the rubber refining process. It can be clearly seen that these three stages have different degrees of influence on the Mooney viscosity, with the kneading stage having the greatest influence. Meanwhile, as the number of epochs increases, FusFormer’s analysis of time steps importance is more accurate. When FusFormer completes training, combined with prior knowledge of tire manufacturing, the significance of these three components highly matches the features of the actual rubber refining process, demonstrating the effectiveness of the interpretable relative multi-head attention.

## 6. Conclusions

In this work, we present FusFormer, a transformer-based model for forecasting time series data. FusFormer makes use of specialized components to properly deal with time series input and static variables across the Mooney viscosity data set. Specifically, these include the following: (1) the time series data-processing scheme, which learns the local context information and encodes the position of the time series; (2) the temporal encoder–decoder framework, which captures relative position information of the input time series and incorporates relative position representations in the self-attention mechanism; (3) the static enrichment module, which introduces GLU to suppress the irrelevant information and control the extent of nonlinear processing. As manifested in the Mooney viscosity prediction case, we show that FusFormer achieves state-of-the-art forecasting performance compared with benchmarks. The benefits of each of our proposed architectural contributions are quantified through a detailed ablation study. Lastly, we introduce the interpretable relative multi-head attention to provide interpretability for the FusFormer. In the interpretability use case, the model can visualize attention weight patterns of time series.

Moreover, our future research plans will be primarily focused in two directions. One is to strengthen the interpretability of the model. We intend to enhance the model’s capability so that it can analyze the importance of different types of inputs and identify significant regime changes. On the other hand, our approach can be further extended to model spatio-temporal data indexed by both time and location coordinates [43]. Attention mechanisms can be generalized to learn relations between two arbitrary points in spatio-temporal space. These two research directions will be pursued in the future.

## Figures and Tables

**Figure 1 entropy-24-00528-f001:**
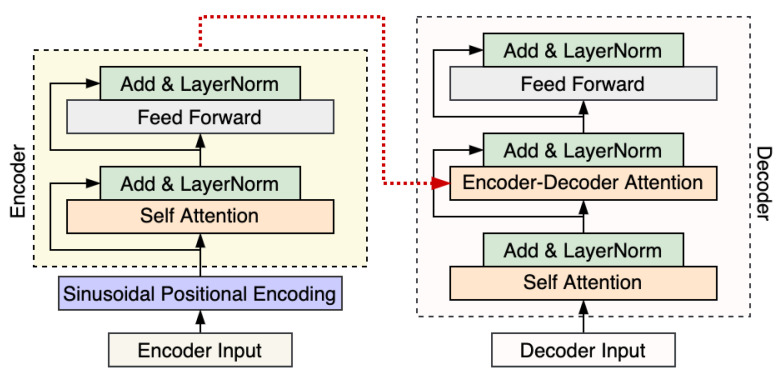
Framework of transformer [16].

**Figure 2 entropy-24-00528-f002:**
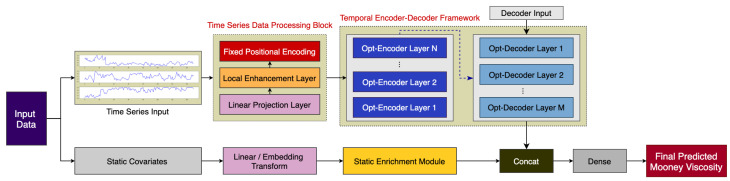
Architecture of FusFormer.

**Figure 3 entropy-24-00528-f003:**
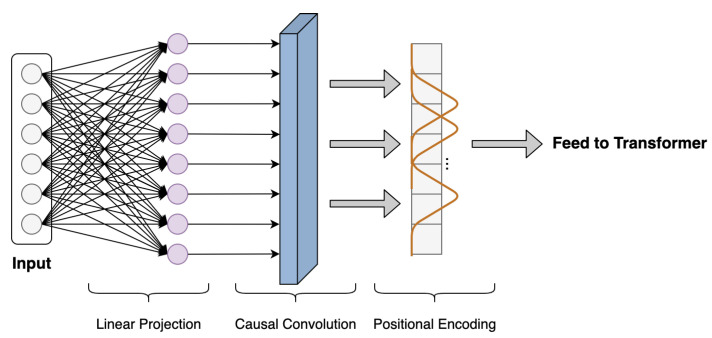
Time series data-processing scheme, including the three components of linear projection, local feature extraction, and positional encoding.

**Figure 4 entropy-24-00528-f004:**
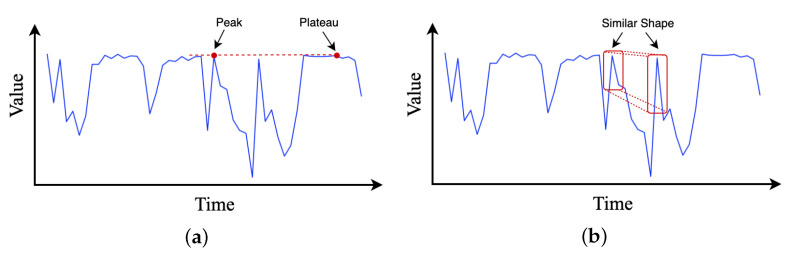
Comparison of the effect of applying convolution to time series input. The red dots in (**a**) focus on the dot close to their value, ignoring the context information, while applying convolution can help the attention mechanism capture the similarity between intervals (**b**) (red block) of time series.

**Figure 5 entropy-24-00528-f005:**
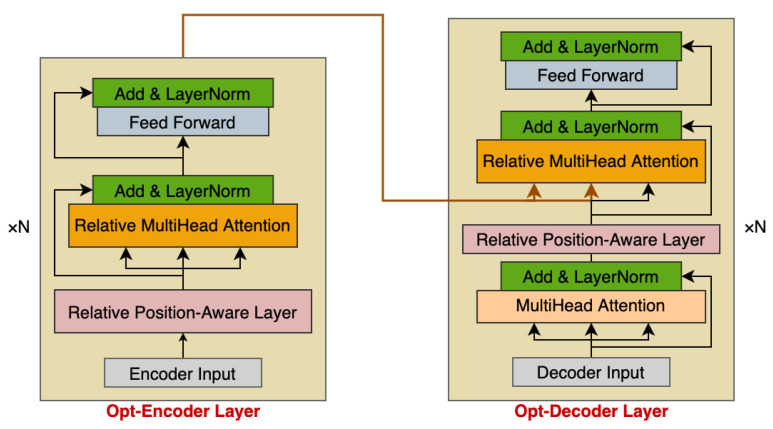
Temporal encoder–decoder framework of FusFormer, including encoder and decoder.

**Figure 6 entropy-24-00528-f006:**
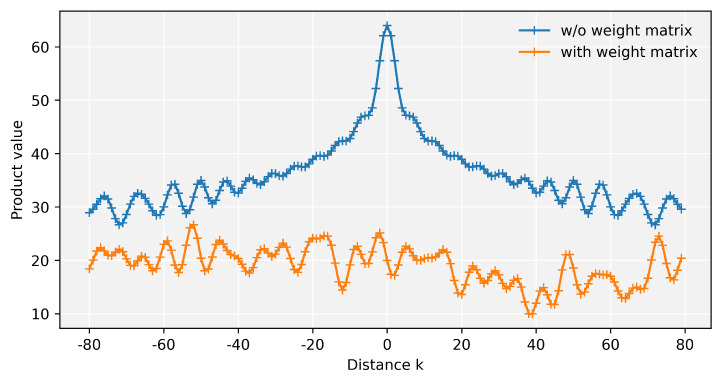
Product value curves. The indigo curve with the clear pattern excludes the weight matrix, and the orange curve represents the weight matrix involved in the calculation.

**Figure 7 entropy-24-00528-f007:**
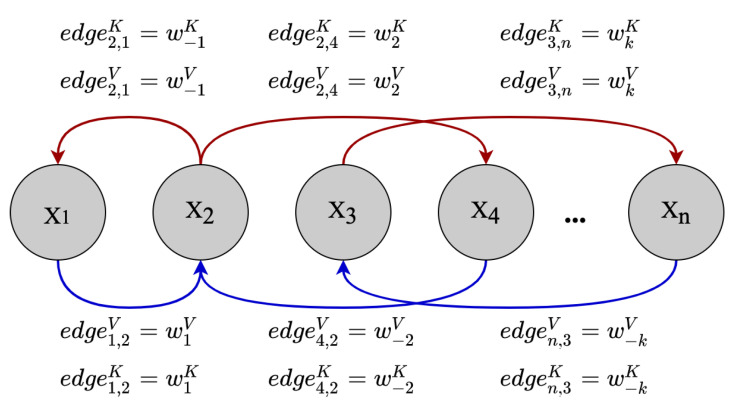
Example edges representing relative positions. We learn representations for each relative position within a clipping distance *k*. The figure assumes 2<=k<=n − 3.

**Figure 8 entropy-24-00528-f008:**
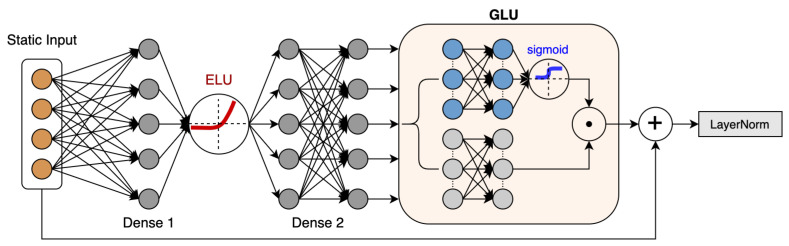
Time series data-processing scheme.

**Figure 9 entropy-24-00528-f009:**
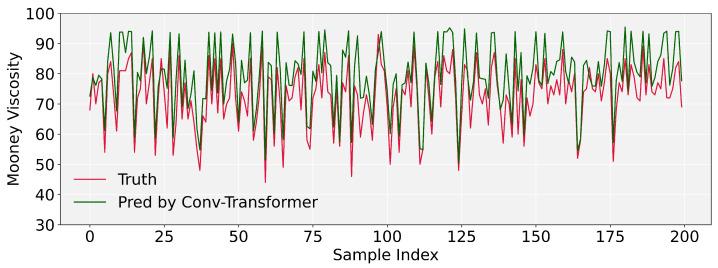
Mooney viscosity prediction results of conv-transformer.

**Figure 10 entropy-24-00528-f010:**
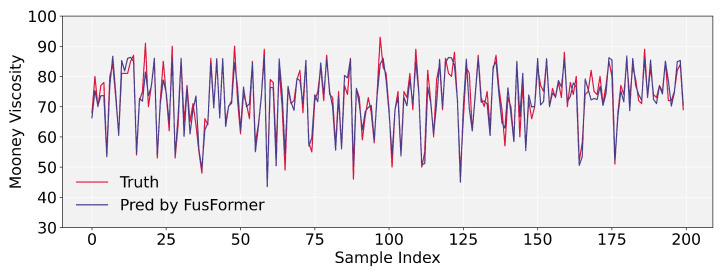
Mooney viscosity prediction results of FusFormer.

**Figure 11 entropy-24-00528-f011:**
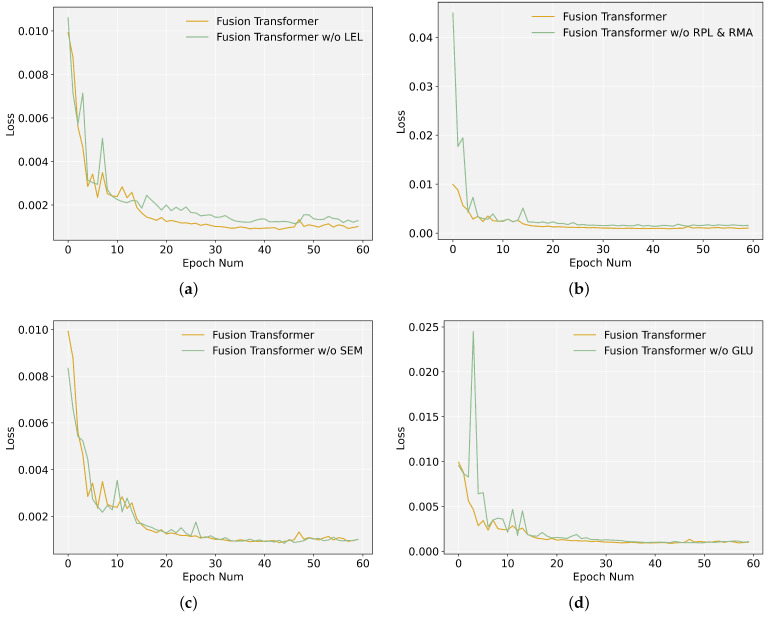
Comparison of loss function curves before and after deleting various modules. (**a**) Ablation analysis of LEL; (**b**) Ablation analysis of RPL and RMA; (**c**) Ablation analysis of SEM; (**d**) Ablation analysis of GLU.

**Figure 12 entropy-24-00528-f012:**
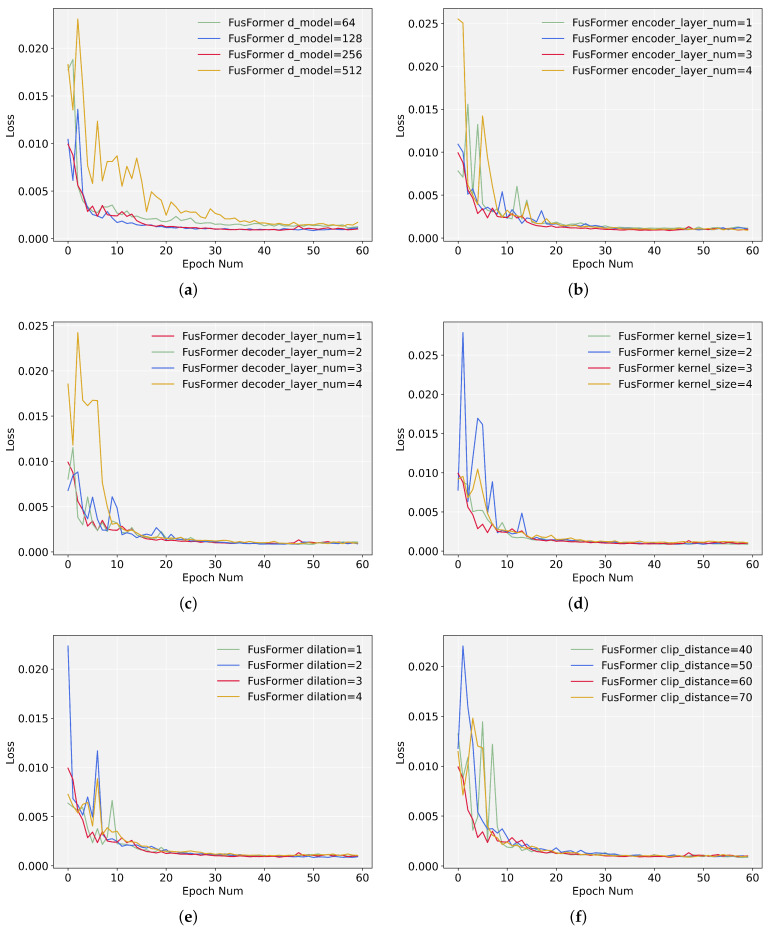
Loss function curves of different values of hyperparameters. (**a**) Loss curves for different values of model dimension; (**b**) Loss curves for different values of encoder layer num; (**c**) Loss curves for different values of decoder layer num; (**d**) Loss curves for different values of kernel size; (**e**) Loss curves for different values of dilation; (**f**) Loss curves for different values of clip distance.

**Figure 13 entropy-24-00528-f013:**
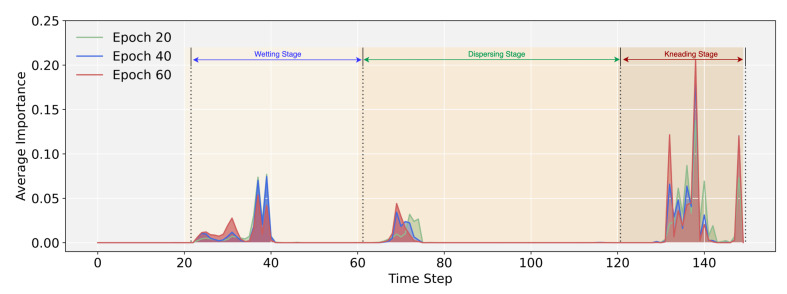
Average attention weight of time series input.

**Table 1 entropy-24-00528-t001:** Optimal hyperparameters configuration.

Optimal Model Hyperparameters				
*Model dimension*	*Encoder layer num*	*Decoder layer num*	*Dropout rate*	*Batch size*
256	3	1	0.05	64
*Learning rate*	*Kernel size*	*Dilation Factor*	*Clip distance*	*Epoch*
0.0001	3	3	60	60

**Table 2 entropy-24-00528-t002:** Summary of model prediction performances with Relative RMSE change.

Model	LSTM	LightGBM	ResNet1D	Transformer	Conv-Transformer	FusFormer
RMSE	6.67	4.56	4.51	4.36	4.23	**3.37**
RMSE Decrease	0%	−31.63%	−32.38%	−34.63%	−36.58%	**−49.32%**
RRSE	0.625	0.445	0.431	0.406	0.394	**0.321**
RRSE Decrease	0%	−28.80%	−31.04%	−35.06%	−36.96%	**−47.68%**

**Table 3 entropy-24-00528-t003:** Comparison of prediction results before and after ablation.

Ablation Module	Model	RMSE	RMSE Decrease
	Transformer	4.36	0%
LEL	FusFormer w/o LEL	3.81	−12.61%
	FusFormer	3.37	−22.48%
	Transformer	4.36	0%
RPL&RMA	FusFormer w/o RPL and RMA	3.66	−16.06%
	FusFormer	3.37	−22.48%
	Transformer	4.36	0%
SEM	FusFormer w/o SEM	3.62	−16.97%
	FusFormer	3.37	−22.48%
	Transformer	4.36	0%
GLU	FusFormer w/o GLU	3.45	−20.87%
	FusFormer	3.37	−22.48%

**Table 4 entropy-24-00528-t004:** Sensitivity analysis of hyperparameters. Bold values indicate optimal tuning results.

Hyperparameter	Set Value	RMSE	Relative RMSE Increase
*Model dimension*	64	3.62	+7.42%
	128	3.54	+5.04%
	**256**	**3.37**	**0%**
	512	3.75	+11.27%
*Encoder layer num*	1	3.61	+7.12%
	2	3.43	+1.78%
	**3**	**3.37**	**0%**
	4	3.54	+5.04%
*Decoder layer num*	**1**	**3.37**	**0%**
	2	3.46	+2.67%
	3	3.63	+7.72%
	4	3.57	+5.93%
*Kernel size*	1	3.47	+2.97%
	2	3.57	+5.93%
	**3**	**3.37**	**0%**
	4	3.51	+4.15%
*Dilation*	1	3.46	+2.67%
	2	3.59	+6.52%
	**3**	**3.37**	**0%**
	4	3.52	+4.45%
*Clip distance*	40	3.49	+3.56%
	50	3.44	+2.08%
	**60**	**3.37**	**0%**
	70	3.42	+1.48%

## Data Availability

Not applicable.

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
