# Peer review of "A Fusion Transformer for Multivariable Time Series Forecasting: The Mooney Viscosity Prediction Case"

_entropy, 2022, doi:10.3390/e24040528_

Round 1

Reviewer 1 Report

The authors propose a transformer-based model, FusFormer, that can be applied to the practical task of multivariable times series forecasting. It is shown that the model outperforms existing forecasting approaches. The case study introduces Mooney viscosity, a parameter that is tracked during the process of rubber refining and which is generally less known than most of the time series used to illustrate similar models. That makes the work more interesting.

The paper is very well written, interesting, each step is clearly explained, each expression is elaborated.

The comments are of technical nature.

The sentence: “Transformer [14] is a novel encoder-decoder model based on the attention mechanism and totally removes recurrent neural networks, which can compute the sequence effectively.” Can RNN that is removed compute the sequence effectively, or a novel encoder-decoder model can compute the sequence effectively?

Sometimes it is written FusFormer, sometimes Fusformer.

Sometimes a space is missing (e.g. “Fig3”).

Define Q, K, V.

Sometimes a “-“ is missing in “self attention”.

There is a reference to "Fig. ??".

In line 230, there is a reference “As illustrated in Fig 10(b) …”. Considering the position of the reference and the context of the entire sentence, 10(b) is not a correct Figure reference. Besides, Fig. 10 is a single figure, it is not split into parts (a) and (b).

Remove indent in line 286 (and everywhere it accidentally occurs).

“We” should not be written with a capital letter, unless positioned at the beginning of the sentence (and in line 438 it is not).

Sometimes it is written “wo” and sometimes “W/O”. I suppose that both stand for “without”. If so, it should be explained and the notation equalized.

In line 583 it is written, “We We intend …”

Figure 4. is puzzling. Please, explain it more clearly.

Figure captions are too laconic. From the captions, the reader should be fully informed of what is in the figure, without the necessity to browse the text.

Fonts in Figures 12, 13, and 14 are too small (axis titles, tick labels, and, to a lesser extent, legends). The fonts should be adjusted to the same size as in Fig. 6.

Line 26: “They incorporate prior knowledge about time series structures such as trend, seasonality and so on,” – the expression “and so on” does not sound professional, especially in the introduction.

Reviewer 2 Report

The authors present FusFormer, a Transformer-based model to forecast time series data with the use of Mooney viscosity prediction as a case study. Performance analysis was carried out to show the effectiveness of the model in producing state-of-the-art forecasting results. This is an interesting work with some contributions. However, there are few issues that need clarification.

  1. Although this work is interesting, it has limited contribution on a very investigated subject. One suggestion I have for the authors relates to the abstract of the paper. It would be better to include some more specific comments about the experimental study in the abstract, so that potential readers of the article can infer about the actual results other than the solution used.
  2. The paper has few grammatical issues, please proofread again before final submission. For example, line 556, what is fig???. Another example, line 583 in the conclusion section, delete the repeated word “We”
  3. Are the datasets publicly available? Any link to visualise the data?
  4. The authors should include root relative squared error (RRSE) to their evaluation criteria to have more comparisons
  5. Why is the model efficiency (s/epoch denoting the time required for each epoch in secs) not calculated for all methods?

Some relevant references are missing such as 

  • Zheng, Shuihua, et al. "Robust soft sensor with deep kernel learning for quality prediction in rubber mixing processes." Sensors 20.3 (2020): 695.
  • Wan, Renzhuo, et al. "Multivariate temporal convolutional network: A deep neural networks approach for multivariate time series forecasting." Electronics 8.8 (2019): 876.

Reviewer 3 Report

Review of "A Fusion Transformer for Multivariable Time Series Forecasting: The Mooney Viscosity Prediction Case" by Yang and Lu (2022)

Authors proposed an interesting method for multivariate time series prediction based on Mooney Viscosity method. The methodology is, in general, well described and results are well illustrated. The work has merit to be published in Entropy journal. However, I have some questions and suggestions to be addressed by the authors:

Main comments:

1. About references, in L25: ", and exponential smoothing [7], and Kalman Filter (Arellano-Valle et al., 2019)". See also about this in L130. In L36: use Khedhiri (2022). References:

Arellano-Valle, R.B., Contreras-Reyes, J.E., Quintero, F.O.L., Valdevenito, A. (2019). A skew-normal dynamic linear model and bayesian forecasting. Computational Statistics 34(3), 1055-1085.

Khedhiri, S. (2022). Comparison of SARFIMA and LSTM methods to model and to forecast Canadian temperature. Regional Statistics 12, 1-19.

2. About the structure, authors used section 1 for Introduction. However, the next section not have numbers. "Related work" (L120) could be the section 2. "Background" (L157) could be section 3. "Experiment" (L370) could be section 4. "Results and conclusions" (L437) could be section 5. "Conclusion" (L567) could be merged to "Results and conclusions".
3. Eq. (2): put the equation as fraction and replace 10000 by 10^4.
4. L192: to avoid confusion, these matrices could be denoted as Q, K and V, respectively.
5. Fig. 4: describe in caption the red line and red block.
6. Proof of page 9: take into account that this is a visual proof, and not a mathematical proof.
7. Eqs. (17)-(20): why these terms are in bold?

Minor comments:

1. Title: "Prediction Case" <-> "Method".
2. L1: "a significant" <-> "an important".
3. L6-L11: this sentence is too large. Please rephrase to be more concise.
4. L22-L23: merge paragraphs.
5. L127: "[23,24]".
6. Before Eq. (1), time series could be denoted as x_i, i=1,...,t_N.
7. For all figures, the caption could be centered.
8. L289: "information."

Round 2

Reviewer 2 Report

All of the reviewer comments have been addressed by the authors.

Reviewer 3 Report

In this 2nd review, I can see that authors have well adressed my previous comments/suggestions. I don't have further comments.